# Brainstem and Cortical Spreading Depolarization in a Closed Head Injury Rat Model

**DOI:** 10.3390/ijms222111642

**Published:** 2021-10-28

**Authors:** Refat Aboghazleh, Ellen Parker, Lynn T. Yang, Daniela Kaufer, Jens P. Dreier, Alon Friedman, Gerben van Hameren

**Affiliations:** 1Department of Medical Neuroscience, Faculty of Medicine and Brain Repair Centre, Dalhousie University, Halifax, NS B3H 4H7, Canada; refat.aboghazleh@dal.ca (R.A.); ellen.parker@dal.ca (E.P.); GvanHameren@dal.ca (G.v.H.); 2Department of Basic Medical Sciences, Faculty of Medicine, Al-Balqa Applied University, Al-Salt 19110, Jordan; 3Faculty of Medicine, Dalhousie University, Halifax, NS B3H 4R2, Canada; 4Department of Integrative Biology, University of California, Berkeley, CA 94720, USA; lynn.yang@berkeley.edu (L.T.Y.); danielak@berkeley.edu (D.K.); 5Center for Stroke Research Berlin, Charité—Universitätsmedizin Berlin, Corporate Member of Freie Universität Berlin, Humboldt-Universität zu Berlin, and Berlin Institute of Health, 10117 Berlin, Germany; jens.dreier@charite.de; 6Department of Experimental Neurology, Charité—Universitätsmedizin Berlin, Corporate Member of Freie Universität Berlin, Humboldt-Universität zu Berlin, and Berlin Institute of Health, 10117 Berlin, Germany; 7Department of Neurology, Charité—Universitätsmedizin Berlin, Corporate Member of Freie Universität Berlin, Humboldt-Universität zu Berlin, and Berlin Institute of Health, 10117 Berlin, Germany; 8Bernstein Center for Computational Neuroscience Berlin, 10115 Berlin, Germany; 9Einstein Center for Neurosciences Berlin, 10117 Berlin, Germany; 10Departments of Physiology and Cell Biology, Cognitive and Brain Sciences, Zlotowski Center for Neuroscience, Ben-Gurion University of the Negev, Beer-Sheva 84105, Israel

**Keywords:** cortical spreading depolarization, electrocorticography, traumatic brain injury, brainstem, oxidative stress

## Abstract

Traumatic brain injury (TBI) is the leading cause of death in young individuals, and is a major health concern that often leads to long-lasting complications. However, the electrophysiological events that occur immediately after traumatic brain injury, and may underlie impact outcomes, have not been fully elucidated. To investigate the electrophysiological events that immediately follow traumatic brain injury, a weight-drop model of traumatic brain injury was used in rats pre-implanted with epidural and intracerebral electrodes. Electrophysiological (near-direct current) recordings and simultaneous alternating current recordings of brain activity were started within seconds following impact. Cortical spreading depolarization (SD) and SD-induced spreading depression occurred in approximately 50% of mild and severe impacts. SD was recorded within three minutes after injury in either one or both brain hemispheres. Electrographic seizures were rare. While both TBI- and electrically induced SDs resulted in elevated oxidative stress, TBI-exposed brains showed a reduced antioxidant defense. In severe TBI, brainstem SD could be recorded in addition to cortical SD, but this did not lead to the death of the animals. Severe impact, however, led to immediate death in 24% of animals, and was electrocorticographically characterized by non-spreading depression (NSD) of activity followed by terminal SD in both cortex and brainstem.

## 1. Introduction

Traumatic brain injury (TBI) is a major global health burden that has been estimated to affect over 40 million people annually [1,2,3]. In most cases of mild TBI—also known as concussion—structural imaging reveals no pathology [4], and transient functional disturbances are thought to underlie the symptoms [4,5]. In contrast, structural brain damages such as intracranial hematomas [6] or diffuse axonal injury [7] are common in moderate-to-severe TBI, with a significant risk of death [8]. In addition, TBI—particularly when severe or repetitive—has been linked to a higher risk of delayed neuropsychiatric complications, including depression [9], epilepsy [10,11], and neurodegenerative diseases such as chronic traumatic encephalopathy [12], Parkinson’s disease [13], and Alzheimer’s disease [14].

Despite the high prevalence and potentially devastating consequences of TBI, the neural network changes underlying the continuum of symptoms from mild to severe TBI are only partially understood [15]. The two main neural network functional disturbances linked with acute brain dysfunction after TBI are epileptic seizures [16,17] and spreading depolarization (SD) [18,19,20]. While both phenomena relate to a sustained depolarization in a large set of neuronal populations, and both potentially propagate along the injured cortex, there are critical differences between the two. During epileptic seizures, sustained depolarization is associated with repetitive firing of action potentials, the transmembrane ion concentration gradients are largely preserved, and the recorded negative direct current (DC) shift is relatively small [21]. In contrast, during SD, neurons depolarize to a level beyond the inactivation threshold of the action-potential-generating channels. The negative DC shift of SD results from the extracellular loss of cations—such as Na^+^ and Ca^2+^ ions—which is not fully compensated by an extracellular gain of K^+^ ions. At the same time, the negative DC shift is an excellent marker of neuronal water influx (cytotoxic edema), because the net cation influx entrains water influx from the extracellular into the intracellular space [21,22,23]. As a consequence of SD-induced depolarization block [24], SD is often associated with a decrease in spontaneous brain activity (spreading depression), which is observed in the alternating current (AC) range of the electrocorticograph (ECoG) above ~0.5 Hz [25] as a rapidly developing reduction in the amplitudes of spontaneous activity, which spreads together with SD between adjacent recording sites. However, in the presence of a severe deficiency of oxidative substrates due to focal or global ischemia [26], SD can occur in the complete absence of any spontaneous brain activity [27]. In such cases, in which activity depression precedes SD, activity depression typically occurs simultaneously at different points in the neural network, and is therefore referred to as non-spreading depression (NSD) of activity. From the initial phase of NSD to the onset of SD, neurons are hyperpolarized, in stark contrast to SD-induced spreading depression, during which neurons are depolarized.

In sedated human patients, SDs were recorded more frequently compared to seizures (usually hours to days) after severe brain insults—including TBI [20,28], subarachnoid hemorrhage [20,29,30,31], and malignant stroke [32,33]—and were found to be associated with worse clinical outcomes [30,34]. In these clinical conditions, seizures and, to an even greater extent, SDs can induce inverse neurovascular responses in terms of severe vasoconstriction, causing secondary deficiency of oxidative substrates and increasing the risk of developing neuronal damage [32,35,36,37]. SDs occur not only in the cortex, but also in many gray matter structures, including the brainstem [38,39,40]. In mice with epilepsy, they were recorded in the brainstem in the context of sudden unexpected death [41]. 

Two previous studies suggested the occurrence of seizures and SDs in models of mild TBI in mice, based on blood flow measurements [42] and electrophysiological recordings [43]. In the present study, we recorded electrophysiological changes from the cortical surface within seconds following mild and severe TBI in rats. We show that SDs are the earliest and most common electrophysiological events following mild and severe TBI, and that seizures are rare. Since oxidative stress is a well-known mediator of injury, we tested the effect of SDs on reactive oxygen species (ROS) formation and antioxidant defense. Repetitive mild TBI resulted in a compromised antioxidant status, leading to impaired defense against ROS and increased ROS in brain tissue following TBI or SDs. 

Since brainstem SDs were reported to be associated with sudden unexpected death [41], we measured cortical and brainstem activity immediately after severe TBI, in which immediate mortality is high. We found that brainstem SDs were rare, but the few detected brainstem SDs were not lethal. However, we recorded cortical and brainstem activity consistent with global ischemia, characterized by NSD of activity followed by terminal SD [26], which has been previously been associated with respiratory arrest and severe fall in systemic blood pressure, leading to early death in rats [44].

## 2. Results

### 2.1. Spreading Depolarization Is Common Following Mild and Severe TBI

We first recorded changes in cortical electrocorticography before and immediately after TBI. Following TBI (Figure 1a), the near-direct current recordings showed the characteristic large slow potential change of SD [45], whereas AC recordings showed the rapidly evolving reduction in amplitudes of spontaneous activity that spread along with SD between adjacent recording sites, which is characteristic of spreading depression of activity (Figure 1b–d) [25,34,43,46]. After mild TBI, SDs were observed after 53% of impacts (*n* = 37 out of 71, Figure 1d), and were recorded within 3 min (124 ± 48 s). A late SD was recorded 43 min after impact in one animal; seizures were rare (*n* = 3; 4%, Figure 1c,d). In sham controls (*n* = 10), cortical activity was characterized by increased amplitude compared to pre-anesthesia baseline, and returned to baseline within less than 5 min (Figure 1e). Neither SDs nor seizures were ever recorded in sham controls. Similar to mild TBI, SDs were detected within minutes (149 ± 43s) after 46% of severe TBI impacts (*n* = 17 out of 37); seizures were uncommon (*n* = 2; 5%). To confirm the propagating pattern of the recorded SDs, changes in cortical surface intrinsic optical signaling (IOS) were measured in parallel with the electrocortical recordings in a subgroup of animals (*n* = 18), using an open cranial window. A slowly propagating change in IOS was measured during spreading depolarization, likely reflecting cytotoxic edema [22,47] and the hemodynamic response to SD [48], confirming the propagating nature of the observed voltage deflection (Figure 1f and Appendix A) [49,50,51].

### 2.2. Oxidative Stress Defense Is Compromised by TBI

SD and the initial, still reversible phase of neuronal cytotoxic edema in the cerebral gray matter are two modalities of the same process [22], but many details of under which circumstances and through which subcellular mechanisms cell damage arises from this in principle reversible process are still unclear, as is the question of whether the same process might also have beneficial effects under certain circumstances [27]. Therefore, we electrically stimulated the frontal cortical surface to elicit SDs [52,53], compared the characteristics of these triggered SDs with TBI-induced SDs, and observed the effects of SD on the brain. Similar to TBI-induced SDs, electrically triggered SDs were detected within three minutes (104 ± 85 s) after stimulation (Figure 2a). The amplitude of electrically triggered SDs (0.26 ± 0.13 mV) was smaller than that of TBI-induced SDs (1.5 ± 0.9 mV; Figure 2b), likely due to the prolonged anesthesia (> 1 h vs. ~5 min), with no difference in duration (Figure 2c). Assuming SD initiation immediately upon stimulation and in proximity to the stimulation site, SD propagation velocity of 4 ± 1 mm/min was measured. No gross structural damage was caused by mild TBI or electrically triggered SDs (Figure 2d–f).

Because oxidative stress has been frequently described after TBI [54,55], we then investigated the potential role of SDs in ROS production. We measured MitoSOX fluorescence in cortical sections from rats after repetitive mild TBI (one hit per day for five consecutive days) (Figure 2g,h). Indeed, MitoSOX fluorescence was significantly higher in TBI-exposed brains (Figure 2j; *n* = 17) compared with sham controls (*n* = 6; Figure 2j), indicating elevated ROS levels. Similar to TBI-exposed brains, MitoSOX fluorescence was higher in brains from electrically induced SDs (*n* = 10; Figure 2i,j) compared to controls (Figure 2j). ROS levels were also higher than controls in animals that endured repetitive mild TBI followed by electrically triggered SDs (*n* = 5; Figure 2j), with no difference compared to naïve animals exposed to SDs. To further assess brain defense capacity from oxidative stress, we measured cortical antioxidant capacity, which is a measurement of the ability of the antioxidant pool present within the cortex to reduce H_2_O_2_ (Appendix A). In naïve control animals, triggering SDs resulted in a higher cortical antioxidant capacity (*n* = 9; Figure 2k) than in non-triggered animals (*n* = 13), likely due to upregulation of antioxidant enzymes in response to oxidative stress (Figure 2g–j). In contrast, repetitive mild TBI-impacted animals failed to increase antioxidant capacity in response to SDs (*n* = 12; Figure 2k), suggesting a compromised antioxidant defense mechanism. Interestingly, measuring mRNA levels of antioxidant enzymes (SOD1, SOD2, and catalase) in the hippocampus—a brain area previously shown to be susceptible to TBI [56,57]—revealed downregulation of SOD1 mRNA expression, with no change in SOD2 or catalase (Figure 2l) after repetitive mild TBI.

### 2.3. Severe TBI-Induced Immediate Death Is Associated with the Typical Electrocorticographic Signature of NSD in Both Cortical Hemispheres and Brainstem, Followed by Terminal SD

SDs previously recorded in the brainstems of transgenic mice with epilepsy were associated with sudden unexpected death [41]. In contrast, in a rat model of severe TBI, early death was attributed to respiratory arrest followed by circulatory arrest [44]. In human patients and rodents, the characteristic electrocorticographic signature of the dying process in the wake of circulatory arrest is NSD of activity followed by terminal SD [27]. We therefore recorded the electrocorticographic events after severe TBI in both cortex and brainstem.

Gross structural damage was observed after severe TBI impacts, manifesting most often as epi-, sub-, and/or intracerebral hemorrhage (Figure 3a). SDs were less frequent in the brainstem than in the cortex (3 of 37 (8%) versus 17 of 37 (46%) (chi-squared test, *p* < 0.001, Figure 3e)). Of note, in these three rats, transient SD in the brainstem (Figure 3b) did not lead to the death of the animals (Figure 3e).

More frequently, following 11 of 37 impacts (~30%), non-spreading depression of brain activity was recorded after the impact, identified by the simultaneous depression of activity in both cortical hemispheres and the brainstem (Figure 3c,d). NSD was typically detected within two minutes (47 s ± 40 s) after the impact, except in one animal, when it was recorded 7 min after the impact. Following NSD, brain activity recovered in two animals who survived the impact (Figure 3c,e); however, in nine animals, NSD of activity was associated with respiratory slowing or complete arrest, and followed by terminal SD recorded in both cortices and the brainstem (Figure 3d,e). All nine animals did not recover from the trauma, and died. The delay between onset of NSD and onset of terminal SD was 125 ± 77 s (*n* = 9, 24%, Figure 3d,e). Sham controls or animals undergoing mild TBI never died, and neither NSD nor terminal SD were ever recorded in these animals (Figure 3e).

## 3. Discussion

Our findings using near-direct current recordings, alternating current recordings, and analysis of IOS show that SD and SD-induced spreading depression are common in both mild and severe TBI, and are initiated within minutes after impact. Our observations build on previous studies showing that direct impact to the dura mater of anesthetized mice results in a bilateral decrease in regional cerebral blood flow (rCBF), which has been suggested to result from SD [18]. Using electrocorticography, which is the gold standard for identifying SDs in both animals and patients [25,34], we confirmed that SDs are commonly triggered by closed head injury. TBI-induced SDs are higher in amplitude than electrically triggered SDs. While the use of a near-DC amplifier (high-pass filter of 0.02 Hz) distorts the signal and affects measurements of amplitude and duration, the difference in SD amplitude (and not duration) could be the result of a longer duration of anesthesia (>1 h), since the amplitude of electrically triggered SDs was higher when anesthetic duration was shorter (~5 min). A suppressing effect of isoflurane on SD has indeed been reported in previous studies [58].

A key feature of SDs is a slow propagating pattern along the cortical tissue. To confirm the propagating nature of TBI-induced SDs, we analyzed changes in IOS using the open cranial window method. Our observed IOS changes showed that SD is indeed associated with a slowly propagating change in cortical IOS [49,50], likely reflecting cytotoxic edema [22,47] and changes in cerebral blood volume due to the hemodynamic response to SD [48]. In contrast to previous reports showing that electrically induced SDs were typically restricted to the ipsilateral hemisphere [59], we recorded TBI-induced SDs in both hemispheres. The bilateral propagation in our recordings suggests multisite initiation, or a different propagation pattern following TBI (Figure 1e and Appendix A).

To what extent the occurrence of SDs affects the outcome of brain injury, and many mechanistic details, are still unknown. Recent human studies confirm that SDs can be frequently recorded in sedated patients hours to days after severe TBI [20,28,60], subarachnoid hemorrhage [45,61], malignant stroke [32,33], intracerebral hematoma [62], and subdural hematoma [63]. Under these conditions, it has been shown that SDs may be associated with an inverse neurovascular response, progression of ischemic injury, and worsened clinical outcomes [30,34]. While blood flow measurements were not carried out in the present study, we show that triggering SD causes elevated levels of reactive oxygen species, which might be due to mitochondrial fragmentation induced by cytotoxic edema [23], and may result in cellular injury or death if not resolved by antioxidants [55]. Indeed, TBI-induced SD has been shown to be associated with apoptotic cell death [42]. Interestingly, while in naïve animals repetitive SDs result in elevated antioxidant capacity within the treated cortex, such a defense mechanism does not occur in TBI-impacted animals. These results suggest a reduced capacity of the injured brain to adapt its antioxidant defense in response to an SD challenge. By measuring the mRNA expression of mitochondrial enzymes, we found that TBI induces downregulation of SOD1—one of the key antioxidant enzymes—in the hippocampus. Hippocampal susceptibility to TBI has been shown previously [56,57], and we observed a neuroinflammatory response to mild injury within the hippocampus (Appendix A). Although cortical mRNA expression was not measured from these same animals, this opens the possibility that TBI-induced weakening of the antioxidant defense may occur via downregulation of this antioxidant enzyme. A decrease in SOD1 expression was also reported after a single mild TBI in mice, along with decreases in SOD2 and glutathione peroxidase expression [64]. In addition, TBI has been reported to exacerbate the ALS phenotype in some mutant SOD1 models [65], and increasing SOD1 function via stem cell transplantation has been shown to be a promising method for faster recovery following TBI [66].

In addition to injury to the cortex via oxidative stress, SD has previously been linked to acutely harmful and dangerous consequences, including death, when it spreads into the brainstem. We therefore measured brainstem electrical activity immediately after severe TBI, with a high prevalence of post-traumatic death. Transient SDs in the brainstem, causing spreading depression of brainstem activity, were recorded after a few severe TBI impacts, but were always nonlethal; this is consistent with previous works showing that recovery from brainstem SD is possible, provided that it is short-lasting [40,41,67]. In other words, we did not find SD in the brainstem to be a causative mechanism for respiratory–circulatory arrest [68]. However, severe TBI could also result in sudden NSD of electrical activity, occurring simultaneously in both cortical hemispheres and the brainstem. When NSD of brainstem activity was short-lasting and not followed by SD, animals recovered and survived the impact. In contrast, animals with prolonged NSD followed by SD did not recover from the impact, and died. Takahashi et al. previously showed that severe TBI in mice and rats can result in sudden apnea and bradycardia with hypertension followed by hypotension, which was associated with NSD, followed by marked elevation in the extracellular potassium concentration [44], and death of the animal. Our findings show good agreement with these observations. The electrocorticography changes we observed are assumed to represent a fundamental pattern that occurs in a characteristic manner in both respiratory and circulatory arrest, and may thus represent anoxic episodes [69], although some variation in detail is possible [70]. We observed NSD as soon as 20 s after the injury. The most likely scenario is that the mechanical impact led directly to brain hypoxia (potentially due to respiratory circulatory arrest) via an ultimately unclear shock mechanism, which led to NSD with the usual latency.

Terminal SDs occur in severe noxious conditions—such as cardiac arrest, severe prolonged hypoxia, hypoglycemia, and tissue exposed to very high concentrations of K⁺ [26,27,71]—but have not been sufficiently shown immediately after acute TBI. We recorded terminal SDs ~125 s after the onset of depression of brain activity, which is consistent with the delay (98 ± 88s) previously observed in patients with TBI, aSAH, and malignant hemispheric stroke during the dying process in the wake of circulatory arrest [26]. In addition, we clearly demonstrated that terminal SDs did not occur simultaneously in both hemispheres and in the brainstem, but that their onsets had distinct temporal latencies (Figure 3c). A terminal SD has two components: on the one hand, there is the initial, still reversible SD component and, on the other hand, a late potential component, which is termed negative ultraslow potential (NUP) [26]. We were not able to detect the NUP in our recordings because we only performed near-DC recordings, and not full-band recordings. In a future study, it would be interesting to find out whether rapid-onset artificial ventilation can prevent the death of the animals, the specific cellular mechanisms that lead to respiratory arrest, and whether they are reversible or not. In contrast, preventing terminal SD in the presence of continued respiratory and circulatory arrest is unlikely to be possible.

In summary, we presented the immediate electrophysiological changes after TBI, showing that both mild and severe TBI frequently lead to SD and SD-induced spreading depression of cortical activity, while seizures are rare. We suggest that SD causes oxidative damage. Importantly, triggering SDs, on the other hand, resulted in higher cortical antioxidant capacity, likely due to upregulation of antioxidant enzymes in response to oxidative stress, but TBI-impacted animals failed to increase antioxidant capacity in response to SDs. This merits further study. We also observed that brainstem SDs inducing spreading depression could spontaneously occur in severe TBI, but were nonlethal. On the other hand, NSD, predominantly followed by terminal SD, indicated the dying process. Further studies are needed to explore whether—and up to which point in the NSD and SD process—artificial ventilation can still lead to survival of the animal.

## 4. Materials and Methods

### 4.1. Animals

Young (10–12 weeks old) adult male Sprague Dawley rats (Charles River, Montréal, QC, Canada, *n* = 146) were double-housed in standard cages at the Dalhousie University animal care center (Tupper building) with access to food, water, and shelter ad libitum in accordance with animal care protocols 16-094 (1 October 2016), 17-105 (1 November 2017) and 19-065 (1 June 2019). Rats were exposed to a normal 12:12 light:dark cycle.

### 4.2. Recording of Cortical Activity

The rats (*n* = 81) were implanted under deep isoflurane anesthesia (3% for induction and 1.5% for maintenance). Blood oxygen saturation (SpO_2_) was monitored continuously using a paw clip connected to an animal oximeter pod (ML325, ADInstruments, Colorado Springs, CO, USA) and a PowerLab data acquisition device (PL3508, ADInstruments). A midsagittal incision (2.5 cm) was made to expose the skull, and two holes (2 × 2 mm in diameter) were drilled for screw placement (stainless steel bone screws, Fine Science Tools, 0.86 mm × 4 mm) in either one or both parietal bones (2 mm posterior to the bregma, 2 mm anterior to the lambda, and 3 mm lateral to the sagittal suture, Figure 4). A ground electrode was inserted into the neck’s subcutaneous tissue. Epidural ECoG electrodes were constructed from Teflon-insulated silver wire (280 µm diameter, A-M Systems, Inc., Sequim, WA, USA) and miniature connectors (Ginder Scientific, Napean, ON, Canada, ABS plug, ref. GS09PLG-220). The silver wires of the electrodes were wrapped around the screws and were fixed to the skull using dental cement. In a subset of animals (*n* = 18), a cranial window (4 × 6 mm in diameter, 2 mm posterior to the bregma, 2 mm anterior to the lambda, and 2 mm lateral to the sagittal suture) was made in between the two ECoG electrodes, as described in [72] (Figure 4), and covered with a transparent plastic cover slip. A cylindrical TBI platform (1 cm in diameter and 1.5 cm in height) was formed above the frontal or parietal bones using dental cement (Figure 4), where a weight was dropped transmitting an impact to the brain. For brainstem recordings, two holes were drilled in the occipital bone (12 mm posterior to the bregma, and 3 mm right and left to the midline) (*n* = 10, Figure 4). Two Teflon-insulated silver wires were placed at these holes and inserted at 9 mm depth into the brainstem. 

### 4.3. Traumatic Brain Injury

TBI was induced at 24–72 h after placing recording electrodes using a modified weight-drop model of TBI [52,73]. Briefly, rats were sedated using an induction chamber (3% isoflurane, 2 L/min O_2_) until the toe-pinch reflex was absent. Rats were then placed in the prone position on a sheet of aluminum foil taped to the top of a plastic box (30 × 30 × 20 cm in depth). A metal bolt (1 cm diameter × 10 cm length) was placed on the dental cement TBI platform. A cohort of animals (*n* = 17) received repetitive mild TBI (1 hit per day for 5 consecutive days). Mild TBI was defined by an impact (500–600 g travelling vertically for 0.85 m along a metal guide rail onto the frontal TBI platform; Figure 2e) that does not cause gross structural damage, nor death after a single hit, but does result in a transient reduction in neurological score at 10 min after impact (Appendix A). For severe TBI, the same weight was dropped from 1.00 m above the TBI platform located above the parietal bone, causing gross structural damage (Figure 3a), 20% immediate mortality after a single hit, or 70% mortality after repeated impacts (Appendix A). Following the impact, animals fell through the foil onto a foam pad placed at the bottom of the box, causing a rotation of the head and neck. Sham controls were anesthetized, but did not receive TBI.

### 4.4. Electrocorticographic Recordings

Animals were tethered to an Octal bio amplifier (ML138, ADInstruments, Sydney, Australia) for recording differential epidural ECoG signals. Near- DC recordings were acquired (sampling rate of 1 kHz) with a high-pass filter of 0.02 Hz, a low-pass filter of 100 Hz, and a 60 Hz notch filter. During the light phase of the day cycle, a one-hour baseline recording was acquired in a recording box (60 × 30 × 30 cm). Immediately following TBI, animals were reconnected to the recording system within <30 s. Electrical recordings were continued for 2 h following TBI. Brainstem recordings were limited to animals undergoing severe TBI. ECoG data were analyzed using LabChart software (version 8) and MATLAB.

### 4.5. Electrical Induction of Spreading Depolarizations

For electrical induction of SDs, a window in the right frontal bone—the presumed skull area of maximal impact force—was created instead of a TBI platform (Figure 4). The frontal bone window (2 × 2 mm in diameter, 2 mm lateral and 2 mm anterior to the bregma) was located anterior to the electrodes for parietal ECoG recordings (*n* = 32). In a subset of animals (*n* = 12), a right parietal cranial window was also prepared for cortical surface imaging. Two stainless steel stimulation electrodes (0.5 mm in diameter) were placed at the right frontal bone window for epidural stimulation. SDs were induced under anesthesia (1.5% isoflurane, 0.8 L/min O_2_), as reported in [49,61,74], using electrical stimulation with 20 volts and 20 hertz for 2 s with a pulse duration of 5–20 ms.

### 4.6. Cortical Surface Imaging

Intravital microscopy (Axio Zoom, V16, Zeiss GmbH; Konigsallee, Germany) was performed as described in [72], immediately (<30 s) following impact or electrical stimulation. Images were taken from the surface of the cortex at a rate of one image/sec using a scientific CMOS camera (PCO Edge 5.5 model, PCO-Tech Canada). Changes in intrinsic optical signal (IOS) were analyzed using MATLAB, as reported in [49,74]. 

### 4.7. Histochemistry

Within 1 week following TBI or electrically induced SDs, animals were perfused with 4% paraformaldehyde, brain tissue was collected, and 30 µm thick coronal sections were cut using a microtome (Microm, HM400). The mounted sections were stained with FD cresyl violet solution (FD NeuroTechnologies, PS102-2). Other mounted sections were incubated for 30 min with MitoSOX red (Thermo Fisher, Eugene, OR, USA, ref. M36008) in the dark to detect reactive oxygen species in vitro [75]. Each slide contained 6 evenly spaced sections and 4 regions of interest (2 on each hemisphere) in the forelimb and hindlimb areas of the cortex and parietal cortex, which were analyzed for each section. MitoSOX-stained cells were considered to be high-ROS cells when MitoSOX fluorescence≥median+(median−lowest 5th percentile). The number of MitoSOX-positive cells was counted and divided by the total number of cells present in that region observed via DAPI staining.

### 4.8. Antioxidant Assay

Within 1 h after the last triggered SD with (*n* = 12) or without (*n* = 9) prior TBI, rats were euthanized, and their brains were dissected for antioxidant capacity assays as previously described [76]. Brain tissue of healthy rats was collected as control (*n* = 13). The injured right cortical hemisphere was isolated and homogenized (100 mg/mL) in 1.55 M KCl. Brain homogenates were added to a reaction mix with myoglobin (Sigma-Aldrich, St. Louis, MO, USA, ref. M0630) and 2,2′-azino-bis(3-ethylbenzthiazoline-6-sulfonic acid) (ABTS) (Roche, Mannheim, Germany, ref. 10102946001), and ABTS oxidation by H_2_O_2_ was measured spectrophotometrically (SpectraMax M3, Molecular Devices, San Jose, CA, USA) at an absorbance of 734 nm (Appendix A). Time to absorbance shift was measured and compared to a Trolox (an artificial antioxidant) (EMD Millipore, Temecula, CA, USA, ref. 648471) standard line.

### 4.9. Real-Time Quantitative PCR

One week after repetitive mild TBI (*n* = 18) or repetitive isoflurane exposure (*n* = 8), rats were euthanized, and hippocampal brain tissue was collected, frozen in liquid nitrogen, and stored immediately at −70 °C. Total RNA was extracted via homogenization in TRIzol reagent (100 mg/mL), addition of chloroform (200 µL/mL TRIzol), and centrifugation at 12,000 rpm for 15 min at 4 °C. Supernatant was resuspended in isopropyl alcohol (500 µL per 1 mL TRIzol reagent) and centrifuged at 12,000 rpm for 10 min at 4 °C. The resultant RNA pellet was washed with 75% ethanol and dissolved in DNase/RNase-free water. cDNA was obtained from this purified total RNA via reverse transcription. mRNA expression was measured using real-time quantitative PCR (40 cycles; Table 1) using a QuantStudio 6 Flex (Applied Biosystems). Mean threshold cycles were normalized to a reference gene (*18S*), and relative gene expression levels were quantified using the 2-ΔΔCT method. Primer sequences (Table 2) were designed using PrimerBank (Spandidos et al., 2010), NCBI/NIH Primer-BLAST, and GenScript Real-Time PCR (TaqMan) Primer Design, and primer physical characteristics were cross-checked with the Integrated DNA Technologies OligoAnalyzer Tool.

### 4.10. Statistical Analysis

The normality of our data was tested using Shapiro–Wilk tests, and non-parametric tests were used when normality could not be assumed. Differences between two groups were determined using Student’s *t*-test. Differences between more than two groups were determined via one-way ANOVA or Kruskal–Wallis test, with Dunnett’s, Tukey’s, or Dunn’s post-hoc tests. The differences in SD, seizure, or NSD prevalence between groups were determined using Fisher’s exact test. All statistical analyses were performed using IBM SPSS Statistics (version 25.0) and GraphPad Prism (version 8.0).

## Figures and Tables

**Figure 1 ijms-22-11642-f001:**
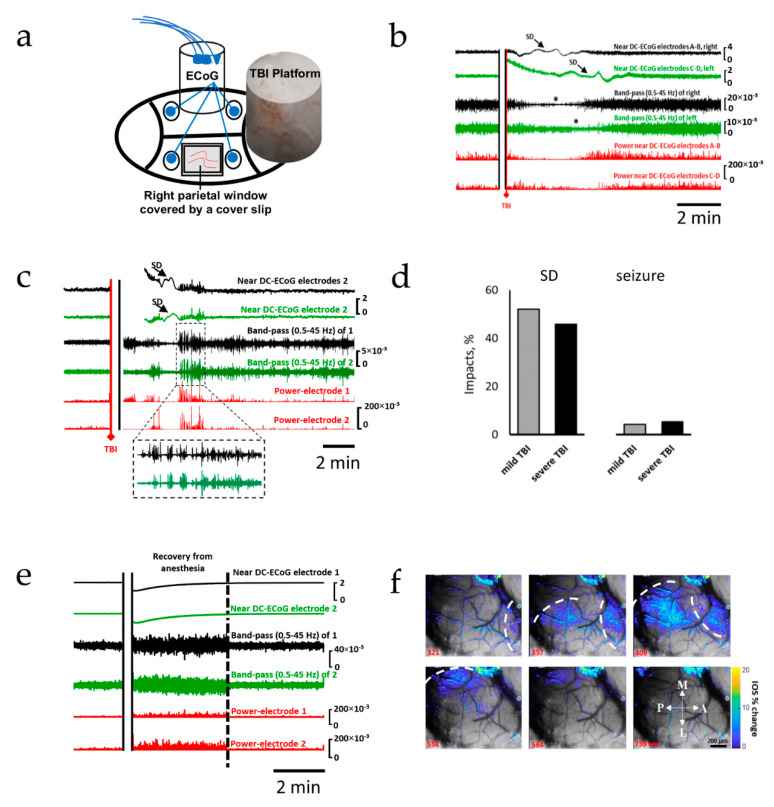
Spreading depolarization is the earliest and most common electrophysiological event after TBI. The upper two traces in each panel show raw ECoG recordings (band-pass: 0.02–100 Hz). The 3rd and 4th traces show band-passed (0.5–45 Hz) activity. Recordings from both the right (black) and left (green) hemispheres are shown. The 5th and 6th traces (red) show the squared activity (power near DC-ECoG). (**a**) Schematic representation of the general experimental setup. (**b**) Recording showing TBI-induced SDs recorded from both hemispheres. (**c**) Recording showing TBI-induced spreading depolarization with seizure activity immediately before and after SD; post-SD seizure is shown with expanded timescale. (**d**) Occurrence rates of SDs and seizures following mild and severe TBI. (**e**) Recording from a non-injured control; recovery from anesthesia is associated with a high-amplitude activity, which returns to pre-impact activity after regain of locomotion (dotted line). (**f**) Intravital microscopy showing changes in intrinsic optical signals during SD; changes in IOS are superimposed onto brain images; SDs propagated medially toward the midline; the dotted line represents the SD fronts. A: anterior; P: posterior; L: lateral; M: medial.

**Figure 2 ijms-22-11642-f002:**
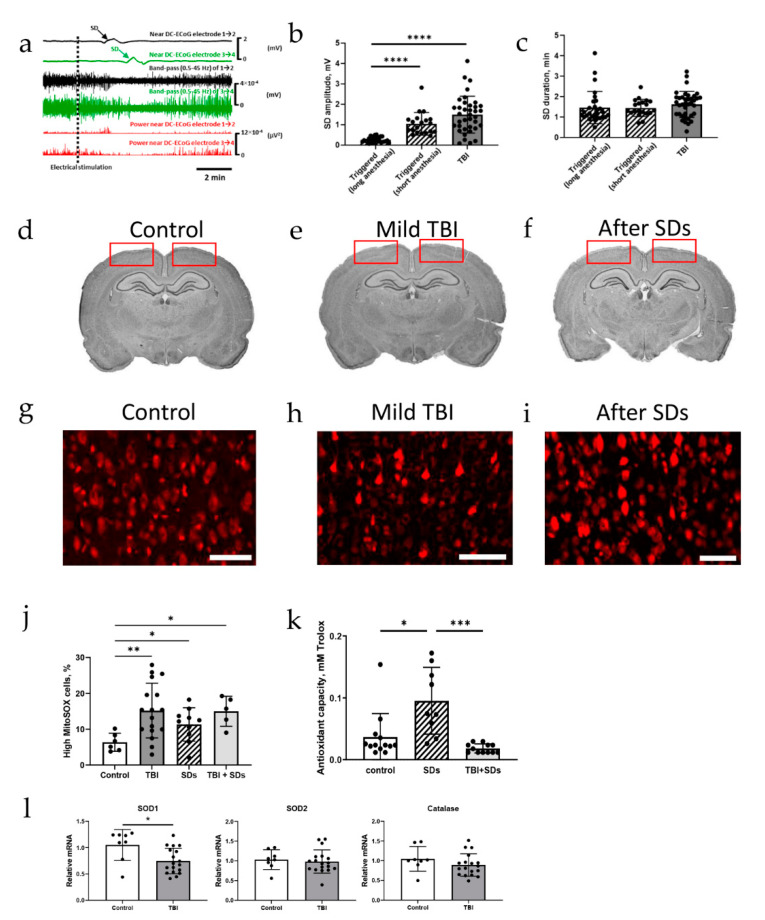
Changes in reactive oxygen species and antioxidant capacity after TBI or spreading depolarizations. (**a**) Recordings from both the right (black) and left (green) hemispheres, and power near DC-ECoG (red) of both hemispheres after electrical stimulation; SDs are triggered in both hemispheres. (**b**) Triggered SDs are lower in amplitude when the rat has been under anesthesia for a longer duration (*p* < 0.0001, Dunn’s test); TBI-induced SDs are higher in amplitude than triggered SDs (*p* < 0.0001, Dunn’s test); each dot represents 1 triggered SD or TBI impact. (**c**) Triggered SDs and TBI-induced SDs are equal in duration. (**d**–**f**) Cresyl staining showing no structural damage following mild TBI or triggered SDs; regions where MitoSOX fluorescence was measured are indicated by the red rectangles. (**g**–**j**) MitoSOX staining results in high-fluorescent-intensity cells in fixed tissue; the percentage of high-MitoSOX cells is higher after mild TBI compared to controls (*p* = 0.0013, Dunnett’s test); SDs are associated with more MitoSOX fluorescence (*p* = 0.044, Dunnett’s test); SDs in TBI-exposed animals also show more MitoSOX fluorescence than controls (*p* = 0.0185, Dunnett’s test), with no difference compared with naïve animals exposed to SDs (*p* = 0.16, Tukey’s test); each dot represents 1 animal. Scale bar = 200 µm. (**k**) Antioxidant capacity increases after triggered SDs (*p* = 0.0189, Dunn’s test), unless triggered SDs were preceded by TBI (*p* = 0.38, Dunn’s test); each dot represents 1 animal. (**l**) SOD1 mRNA expression is decreased in repetitive mild TBI-impacted rats (*p* = 0.0105, unpaired *T*-test), but catalase (*p* = 0.22) and SOD2 (*p* = 0.68) mRNA expression are not affected by mild TBI; each dot represents 1 animal. * *p* < 0.05, ** *p* < 0.01, *** *p* < 0.001, and **** *p* < 0.0001.

**Figure 3 ijms-22-11642-f003:**
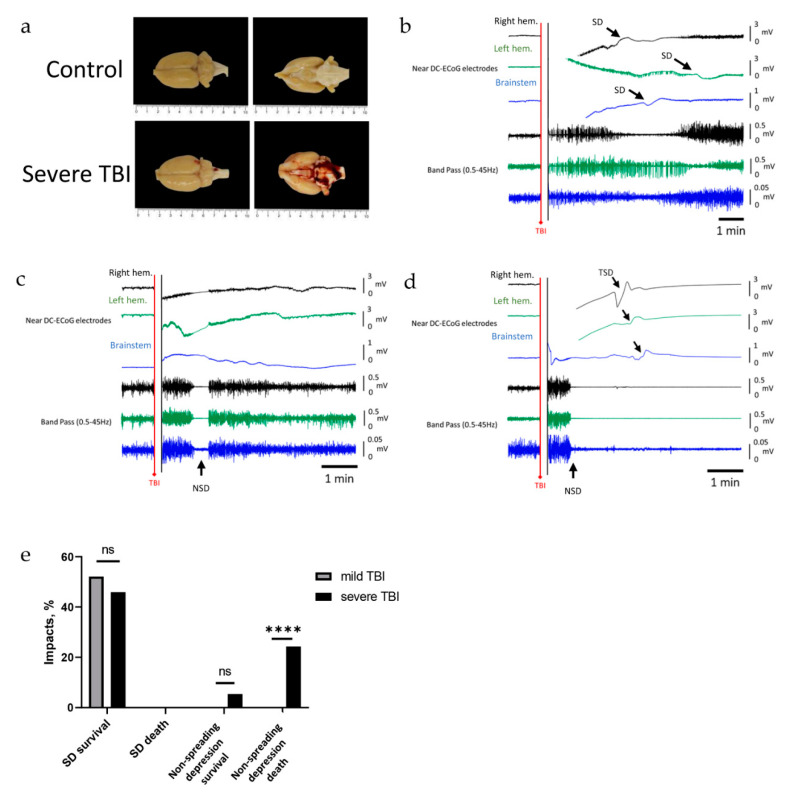
The electrophysiological changes in brain hemispheres and brainstem following severe TBI. The first two traces represent raw ECoG recordings (band-pass: 0.02–100 Hz) from the right (black) and left (green) hemispheres, and the third trace (blue) is from the brainstem. Traces 4, 5, and 6 show brain activity (band-pass 0.5–45 Hz) of the first three ECoG traces, respectively. (**a**) Hemorrhages at the cortex and brainstem observed after severe TBI. (**b**) Recording after severe TBI shows SD and associated depression of activity in both cortical hemispheres and the brainstem. (**c**) Recording showing non-spreading depression of activity occurring simultaneously in both cortical hemispheres and the brainstem; activity returns and the rat survives. (**d**) Recording showing lasting non-spreading depression of activity followed by terminal spreading depolarizations (TSDs, arrows) in both cortical hemispheres and the brainstem, and death. (**e**) Occurrence rate of SDs and non-spreading depression in mild and severe TBI, and their association with TBI outcomes; non-spreading-depression-associated death occurs significantly more frequently (*p* < 0.0001, Fisher’s exact test) in severe TBI. **** *p* < 0.0001.

**Figure 4 ijms-22-11642-f004:**
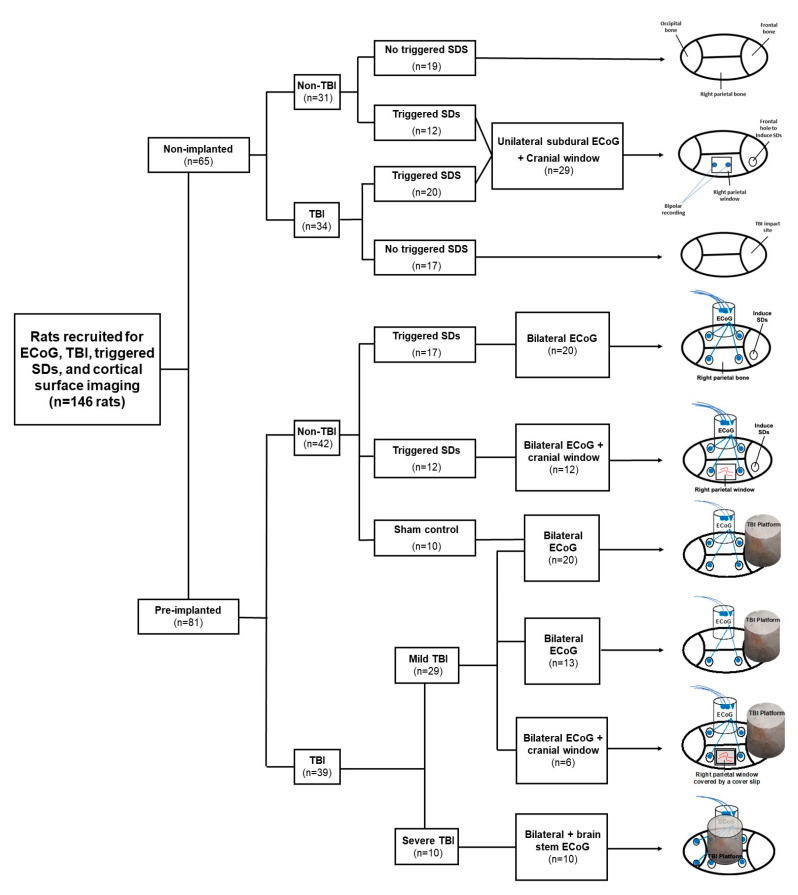
Flowchart of the study design. Schematic drawings show the location of ECoG recording and electrically triggering SDs, right parietal window for cortical monitoring, and constructed platforms for TBI.

**Table 1 ijms-22-11642-t001:** Primer sequences used for RT-qPCR.

Gene	Primers
SOD 1	AAGTGCTGTTGAGTCCAGGT CCATTTCCTCCAGGGTGACT
SOD2	ACTGGTTTCTCAGCTCCTC TCAGGAGCCACAAGTGAGAG
Catalase	ATGGCTCCAAGCGATGTTTC AAGGGTGCTGAATGCCTACT

**Table 2 ijms-22-11642-t002:** Thermocycling conditions for RT-qPCR. Step 3 and 4 were repeated for 40 cycles.

Step	Temperature (°C; Ramp Rate: 1.6 °C/s)	Time (min)
1	50	2:00
2	95	10:00
3	95	0:15
4	60	1:00
5	95	0:15
6	60	1:00
7	95	0:15

## Data Availability

The data presented in this study are available on request from the corresponding author.

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
