# Peer review of "Brainstem and Cortical Spreading Depolarization in a Closed Head Injury Rat Model"

_ijms, 2021, doi:10.3390/ijms222111642_

Round 1

Reviewer 1 Report

In the manuscript, “Brainstem and cortical spreading depolarization in a closed head injury rat model” the authors present evidence of SDs as a prominent feature immediately after experimental TBI and explore relation to ROS and antioxidant status.  This is a timely manuscript, as clinically the evidence for SDs after TBI is strong and additional research is needed to better understand this phenomenon.

Although the SD data is generally well presented, the manuscript lacks cohesiveness and methodological detail. Specific comments are below.

1) The overall rationale and objectives of this study are unclear. The closing paragraph of the introduction (lines 85-98) do not clearly state what the study hopes to show. The connection between antioxidant status, ROS, and brainstem activity is not clear.

2) Figure 1 – please better explain the dotted line in 1(d)

3) Please provide more information about the animals used: where purchased, housing conditions, light dark cycle, etc. Were the animals used for ECoG recordings also used for histochemistry and antioxidant assay?  The total number of animals used in the study does not appear to match Figure 4 or a total animal number of 71. Please state the total number of animals included in the entire study, including animals excluded from analysis due to death or otherwise. 

4) For the TBI model, please provide a reference or supplemental data that shows the characterization of this modified weight-drop model.  The paper referenced by the authors is a model in juvenile rats and no dental cement platform was used.  The characterization of this as a valid TBI model needs to be presented (or appropriately referenced). Were any other characteristics used to characterize the model and define the injury severity such as righting reflex time, neurological severity score? The authors state that mild TBI was “defined by an impact that does not cause death or structural damage” (line 342). What defines moderate-severe TBI?  Please provide a low magnification image of severe TBI histology such that structural damage can be seen / compared between the injury groups. The method for inducing repetitive TBI is not explained sufficiently in the manuscript.  Please provide these details in the methods.

5) Details are missing for how the MitoSOX analysis was performed.  How many sections were analyzed per animal?  Were sections evenly spaced, was a standard region of interest (defined either by anatomical landmarks or approximate stereotactic coordinates) used?

6) Figure 2 (D) and (e) – please clarify the n.  Does the count represent the number of animals or the number of SDs? If the authors wish to make a point about effect of anesthesia on SDs, then details regarding type, percentage, and length of anesthesia should be included (what defines short versus long anesthesia).  Additionally, it is unclear how c-e are related to the rest of the panels in Figure 2.

7) line 146 – is this intended to be a callout for Figure 2c (not Figure 3c)?

8) It would be useful to compare control, SD, TBI alone, and TBI+SDs.  It is unclear why these results are presented in separate analyses in Figure 2(b) and (h).

9) It is unclear why the SOD and catalase analysis did not include a group of animals with SDs.  This data should be added or the rationale explained.

10) Figure 3 – arrows would be useful to illustrate features of the tracings in (a) and (b)

11) The presented data showing the NSD and terminal SD in severe TBI is interesting.  However, it seems removed from the rest of the paper since evaluation of ROS and antioxidant capacity are not included.  Please include this additional analysis. Alternatively, possibly the antioxidant data is out of place and the manuscript could be stronger with a focus on SD characteristics.

12) As this manuscript is focused on cortical and brainstem SDs, it is unclear why hippocampal brain tissue (line 394) was collected for analysis of SOD and catalase.

13) The discussion should better connect the seemingly disconnected pieces of data presented in the manuscript. Also limitations to the data and interpretation of the results should be discussed.

Reviewer 2 Report

1) Please use more conventional panel labelling in figures. Place a, b, c … at the Top left of each panel.

2) Figure 1 would benefit by showing schematic experimental setup with a position of electrodes and the TBI impact site.

3) In general, the manuscript lacks some important quantifications, such as SD characteristics, depression levels of electrographic activity in mild and severe TBI groups, the rate of SD propagation in electrical and IOS recordings.

4) Seizures shown on Figure 1b could be also presented on expanded time scale, is there any lead of population spikes between the electrodes? Please also describe procedures for seizure detection.

5) Panel 1e, please indicate the site of impact. This figure would me more clear if the SD fronts are shown rather than the raw signal.

6) Statistics: as sample size for some variables is relatively small to perform adequate test for normality, it could be suggested to perform non-parametric tests for the group descriptions and comparisons, and to update results and figures accordingly.

7) Figure 3: please add descriptions of conditions and traces directly to the figure (e.g. indicate which trace is cortex and brainstem). In general try to make it easier to the reader to understand the results without reading legends.

Reviewer 3 Report

This is an interesting paper.

I have the following critical remarks:

1. Please indicate more specifically what the 'antioxidant assay' measures. What about the GSH level?

2. How the cell counting for MitoSOX positiv cells was performed? % of what has been plotted (of the cresyl violet posive cells from the adjacent slice)?

3. It is not clear how the transcripts levels for SOD1, catalase and SOD2 were assessed. Which reference gene was used?

Round 2

Reviewer 1 Report

The authors have appropriately addressed the reviewer concerns

Reviewer 3 Report

The authors have addressed all of my concerns.